# Effectiveness of Roux-en-Y Gastric Bypass Versus Hypocaloric Diet in Reducing Cardiovascular Risk Factors in Obese Adults: A Systematic Review and Meta-Analysis

**DOI:** 10.3390/jcm14238349

**Published:** 2025-11-24

**Authors:** Darío S. López-Delgado, Carlos A. Narvaez, Nancy Doris Calzada Gonzales, Rodrigo Martinez-Galaviz, Oriana Rivera-Lozada, Joshuan J. Barboza

**Affiliations:** 1Hospital Universitario Departamental de Nariño, Pasto 520001, Nariño, Colombia; 2Facultad de Ciencias de la Salud, Universidad Nacional Autónoma de Nicaragua, Managua 14172, Nicaragua; 3Facultad de Medicina, Universidad Nacional Hermilio Valdizan, Huánuco 10001, Peru; 4Facultad de Medicina, Universidad Autónoma de San Luis Potosí, San Luis Potosí 78421, Mexico; 5Vicerrectorado de Investigación, Universidad Señor de Sipán, Chiclayo 14002, Peru

**Keywords:** bariatric surgery, hypocaloric diet, cardiovascular risk, obesity, prevention and control

## Abstract

**Background:** Obesity-related dyslipidemia, hypertension, and insulin resistance are major global concerns. Roux-en-Y gastric bypass (RYGB) yields greater weight loss than lifestyle therapy, but its advantage over hypocaloric diets for cardiometabolic risk is uncertain. We compared RYGB with hypocaloric diets in adults with obesity. **Methods:** We included clinical trials and observational studies that enrolled adults (BMI ≥ 30 kg/m^2^) and compared RYGB with hypocaloric diets (500–1000 kcal/day deficit). The primary outcome was LDL-cholesterol; the secondary outcomes were HDL-cholesterol, BMI, systolic blood pressure, triglycerides, and HbA1c. Studies with additional procedures or co-interventions were excluded. PubMed, EMBASE, Scopus, Web of Science, and Google Scholar were searched through 1 July 2025. Risk of bias was assessed with RoB 2.0 and the Newcastle–Ottawa Scale. Random-effects meta-analyses were performed. **Results:** Eight studies (*n* = 622) met the criteria. Versus diet, RYGB did not significantly reduce LDL-cholesterol (MD −9.52 mg/dL; *p* = 0.09). Trends favored RYGB for BMI (MD −4.02 kg/m^2^; *p* = 0.06) and triglycerides (MD −18.58 mg/dL; *p* = 0.06). Changes in HDL (+2.81 mg/dL), systolic blood pressure (−2.77 mmHg), and HbA1c (−0.22%) were small and non-significant. The certainty of evidence was low across outcomes. **Conclusions:** Current evidence does not demonstrate superiority of RYGB over hypocaloric diets for cardiometabolic risk markers. Dietary therapy remains first-line; RYGB may be considered for selected patients.

## 1. Introduction

Global obesity has nearly tripled since 1975; about 2 billion adults live with overweight, and ~650 million live with obesity [1]. Excess adiposity drives dyslipidemia, type 2 diabetes, hypertension, and low-grade inflammation, elevating risks of coronary disease, stroke, and cardiovascular death [2,3]. Current guidelines prioritize lifestyle therapy—hypocaloric diets, physical activity, and behavioral strategies—because these approaches improve metabolic syndrome and inflammatory markers; however, randomized trials seldom show fewer hard coronary events with medical weight loss alone [2,3,4].

Because sustaining weight loss by diet alone is difficult, clinicians increasingly use bariatric surgery; Roux-en-Y gastric bypass (RYGB) ranks among the most common procedures worldwide [5]. Prospective studies show large, durable weight loss and meaningful improvements in blood pressure, glycemia, and lipids versus non-surgical care [4]. Observational cohorts also report lower long-term cardiovascular events and mortality after bariatric surgery compared with non-surgical management [2].

Recent meta-analyses that compare surgery with intensive non-surgical regimens (often pharmacotherapy and structured programs) show substantially greater weight loss and cardiometabolic gains after surgery [6,7,8]. Another review similarly reports superior long-term weight loss and better hypertension and glycemic control versus combined non-surgical interventions [9]. Yet these comparisons pool heterogeneous medical strategies and rarely isolate hypocaloric diets alone. Individual randomized and controlled observational studies directly compare RYGB with prescribed hypocaloric diets and typically find greater weight loss and metabolic benefit after surgery [3,10]. Therefore, we conducted a systematic review and meta-analysis to quantify the comparative effects of RYGB versus hypocaloric diets on cardiovascular risk factors in adults with obesity.

## 2. Methods

### 2.1. Study Design

We conducted this systematic review and meta-analysis in accordance with PRISMA 2020 recommendations [11], as detailed in the PRISMA 2020 checklist (Appendix A), and registered the protocol in PROSPERO (CRD420251061374). We prespecified objectives, eligibility criteria, outcomes, and analytic methods before screening. We used the following PICOT parameters:

Population: Adults ≥ 18 years with BMI ≥ 30 kg/m^2^.

Intervention: Roux-en-Y gastric bypass (RYGB).

Comparator: Hypocaloric diet was defined as a structured calorie-restricted regimen targeting a 500–1000 kcal/day deficit below estimated energy needs, corresponding to a total caloric intake of approximately 800–1200 kcal/day, with the intensive hypocaloric phase typically lasting 4–6 weeks [12].

Outcomes: primary—LDL-cholesterol; secondary—HDL-cholesterol, BMI, systolic blood pressure, triglycerides, and HbA1c.

Timeframe/Designs: Randomized controlled trials and observational comparative studies reporting, or permitting derivation of, mean changes or end values for the outcomes of interest.

### 2.2. Information Sources and Search Strategy

We searched PubMed, Embase, Web of Science, Scopus, and Google Scholar from inception to 1 July 2025, without language restrictions. We combined controlled vocabulary and keywords (e.g., “gastric bypass”, “hypocaloric diet”, “body mass index”, “glycated hemoglobin”, “lipid profile”, “systolic blood pressure”, and synonyms) using Boolean operators AND/OR. We also screened reference lists and forward citations of relevant reports. Full reproducible strategies are provided in the Appendix A.

### 2.3. Eligibility Criteria

The inclusion criteria included the following: randomized or observational comparative studies enrolling adults (≥18 years) with BMI ≥ 30 kg/m^2^ that compared RYGB with a hypocaloric dietary intervention and reported at least one cardiometabolic outcome (LDL, HDL, triglycerides, HbA1c, systolic blood pressure, or BMI).

The exclusion criteria included the following: studies focusing on other bariatric procedures (e.g., sleeve gastrectomy, gastric banding); co-interventions beyond standard perioperative care or diet alone (e.g., GLP-1 agonists, orlistat, structured pharmacologic programs); pediatric-only populations; no direct hypocaloric-diet comparator; or absence of relevant outcomes. We imposed no restrictions by publication year, country, or language. We excluded one potentially eligible study at full-text screening due to the unavailability of the complete article despite reasonable interlibrary and author-contact attempts.

The outcomes included the following: The primary outcome was low-density lipoprotein cholesterol (LDL-C, mg/dL). The secondary outcomes were high-density lipoprotein cholesterol (HDL-C, mg/dL), triglycerides (mg/dL), glycated hemoglobin (HbA1c, %), systolic blood pressure (SBP, mmHg), and body mass index (BMI, kg/m^2^). For all outcomes, lower values were considered beneficial, except for HDL-C, where higher values indicate a favorable change.

### 2.4. Study Selection

Two reviewers independently screened titles/abstracts and then full texts against the prespecified criteria. They resolved disagreements by discussion, with a third reviewer adjudicating when needed.

### 2.5. Data Extraction and Items

Using a piloted form, two reviewers independently extracted study characteristics (author, year, design, setting), participant features, intervention/comparator details (diet prescription, surgical technique), follow-up duration, and outcome data (BMI, HDL, LDL, triglycerides, HbA1c, systolic blood pressure). They resolved discrepancies by consensus. When necessary, we derived effect estimates from reported statistics using standard methods.

### 2.6. Risk of Bias Assessment

Two reviewers independently assessed risk of bias using RoB 2.0 for randomized trials [13] and the Newcastle–Ottawa Scale for observational studies [14]. They reconciled disagreements through discussion or third-party adjudication. We considered risk-of-bias judgments when interpreting the results.

### 2.7. Statistical Analysis

We synthesized continuous outcomes as mean differences (MDs) with 95% confidence intervals using random-effects models (inverse-variance method) and the Paule–Mandel estimator for between-study variance (τ^2^). We quantified heterogeneity with I^2^ (low < 30%, moderate 30–60%, high > 60%). Planned sensitivity analyses included fixed-effect (inverse-variance) models and leave-one-out influence analyses. Where necessary, we converted units to ensure consistency across studies. We performed all analyses in R (v4.5.1) using the meta package (e.g., metacont).

### 2.8. GRADE Assessment

We applied GRADE to rate the certainty of evidence for each outcome across domains of risk of bias, inconsistency, indirectness, imprecision, and publication bias, and we summarized judgments in Summary of Findings tables generated with GRADEpro GDT [15].

### 2.9. Assessment of Small-Study Effects

When ≥7 studies were available for a given outcome, we explored small-study effects using funnel plots and Egger’s regression test. For our primary lipid outcome (LDL-cholesterol), we generated a funnel plot of the mean difference against its standard error and applied Egger’s test, acknowledging that the power of this test is limited when fewer than 10 studies are included.

## 3. Results

### 3.1. Study Selection and Characteristics

We identified 1886 records across PubMed (*n* = 290), Scopus (*n* = 348), Web of Science (*n* = 502), Embase (*n* = 443), and Google Scholar (*n* = 300). After removing 860 duplicates, we screened 1026 titles/abstracts and excluded 1006 that did not meet the inclusion criteria. We sought 20 full texts, could not retrieve 1, assessed 19 in full, and excluded 11 for ineligibility. Ultimately, we included eight studies (total *n* = 622) in both the qualitative and quantitative syntheses (Figure 1).

Table 1 presents core characteristics (design, sample size, intervention/comparator definitions, follow-up), with additional details provided in the Appendix A. Across studies, the hypocaloric comparator consistently prescribed a 500–1000 kcal/day deficit, while the surgical arm used standard RYGB techniques; outcome ascertainment generally followed routine clinical or laboratory protocols.

Follow-up for cardiometabolic outcomes varied across trials. Three studies evaluated short-term effects after 4–9 weeks of intensive caloric restriction or matched weight loss (Eriksson et al. [16], Karlsson et al. [17], Yoshino et al. [22]), whereas four trials followed participants for approximately 12 months (Gómez-Martín et al. [19], Raffaelli et al. [10], Tam et al. [21], and Saiyalam et al. [18], the latter including a 12-week VLCD phase followed by 12-month follow-up). Øvrebø et al. [20] reported longer-term trajectories with follow-up extending up to 5 years, depending on the lifestyle program. Overall, most weight and lipid outcomes in our meta-analysis are based on follow-up between several weeks and 12 months, with limited very long-term data.

### 3.2. Meta-Analytic Findings

#### 3.2.1. Lipid Profile

LDL-C. RYGB did not significantly reduce LDL compared to a hypocaloric diet (MD −9.52 mg/dL; 95% CI −20.55 to 1.50; *p* = 0.09; I^2^ = 93%). Both randomized and observational strata pointed toward lower LDL after RYGB, but the imprecision and wide prediction limits (−43.98 to 24.93) indicate that effects likely vary materially by setting (Figure 2).

HDL-C. We observed a small, non-significant HDL increase with RYGB (MD 2.81 mg/dL; 95% CI −2.83 to 8.45; *p* = 0.33; I^2^ = 96.9%). Subgroup analyses by design did not resolve the heterogeneity (test for subgroup differences *p* = 0.97) (Figure 3).

Triglycerides. The pooled effect favored RYGB with a borderline reduction (MD −18.58 mg/dL; 95% CI −37.71 to 0.54; *p* = 0.057; I^2^ = 70.4%). Directionality was consistent across designs, but the prediction interval (−75.63 to 38.47) indicates substantial dispersion (Figure 4).

#### 3.2.2. Body Mass Index

RYGB produced a larger BMI decrease than diet, narrowly missing statistical significance overall (MD −4.02 kg/m^2^; 95% CI −8.16 to 0.12; *p* = 0.057; I^2^ = 95.5%). The follow-up duration modified the effects: studies with ≥1 year reported a markedly greater reduction (MD −10.55; 95% CI −12.34 to −8.77; I^2^ = 0%), whereas shorter follow-up showed smaller but significant decreases (MD −0.83; 95% CI −1.18 to −0.47; I^2^ = 0%); the subgroup difference was significant (*p* < 0.0001; k = 6; *n* = 233) (Figure 5).

#### 3.2.3. Glycemic Control and Blood Pressure

HbA1c. We found no significant difference between groups (MD −0.22%; 95% CI −1.09 to 0.65; *p* = 0.62; I^2^ = 84.6%; prediction interval −3.13 to 2.69) (Figure 6). Baseline glycemic control and medication use likely contributed to between-study variability.

Systolic blood pressure. The pooled effect modestly favored RYGB (MD −2.77 mmHg; 95% CI −6.39 to 0.85; *p* = 0.13; I^2^ = 27.2%), with relatively consistent findings and a prediction interval of −11.20 to 5.66 mmHg (Figure 7). Although small, this trend aligns with weight-loss-mediated BP improvements.

### 3.3. Adverse Events

Safety reporting was sparse and heterogeneous. Gómez-Martín et al., 2017 reported one postoperative bleed and two minor wound infections after RYGB, with no complications in the diet arm [19]. Øvrebø et al., 2017 noted mild gastrointestinal symptoms (nausea, constipation) in three RYGB patients; none required stopping treatment [20]. Raffaelli et al., 2014 documented one anastomotic leak requiring re-intervention after RYGB [10]. The diet-only groups experienced negligible adverse events. Overall, RYGB carried low but expected surgical risks; inconsistent safety definitions and underreporting limit firm comparative conclusions.

### 3.4. Risk of Bias

Seven observational studies performed well on the Newcastle–Ottawa Scale, showing strong selection processes, appropriate control of key confounders in six studies, and rigorous outcome assessment with adequate follow-up (Appendix A). The single randomized trial showed low risk of bias across RoB 2.0 domains, including randomization, adherence, missing data, outcome measurement, and reporting (Appendix A).

### 3.5. Certainty of Evidence

Using GRADE, we rated certainty very low for all outcomes due to risk of bias, inconsistency (notably high I^2^ values and wide prediction intervals), and imprecision of pooled estimates. These limitations underscore the need for adequately powered, well-controlled trials with standardized diet prescriptions, consistent laboratory methods, and prespecified safety reporting. Detailed judgments appear in Appendix A.

### 3.6. Small-Study Effects/Publication Bias

For LDL-cholesterol (k = 7 trials), visual inspection of the funnel plot (Appendix A) did not suggest marked asymmetry. Egger’s regression test could not be performed due to the small number of included studies (k < 10), so the presence of publication bias cannot be ruled out.

## 4. Discussion

In this systematic review and meta-analysis, between-group differences between RYGB and hypocaloric diets on most cardiovascular risk markers were not statistically significant; however, the direction and magnitude of point estimates consistently favored RYGB—most notably for BMI, triglycerides, LDL cholesterol, HbA1c, and systolic blood pressure—indicating a coherent trend toward clinical benefit that may be masked by imprecision and between-study heterogeneity. In routine practice, RYGB yields ~25–30% total weight loss within 1–2 years versus ~5–10% with diet alone [23], and ≈25% loss can be sustained at a decade, while diet-induced losses commonly recur within a few years [24]. During the initial weight loss period of around 6 to 12 weeks, variations in glycemic levels and blood pressure become comparable between RYGB and very low calorie diets [17]; the divergence emerges with durability—RYGB shows less weight regain and more persistent metabolic benefits over time [25].

Compared with nonsurgical controls, RYGB is associated with sustained weight loss at 12 years and significantly lower incidence and higher remission of type 2 diabetes, hypertension, and dyslipidemia, with improvements in metabolic parameters persisting over the long term [26]. Among patients with type 2 diabetes, RYGB is further linked to lower risks of microvascular and macrovascular complications (including MACE and nephropathy) over ≈11 years of follow-up [27]. At one year, RYGB also produces larger triglyceride decreases, higher HDL-C and apolipoprotein A4, and improvements in hepatic insulin resistance versus diet alone, with parallel reductions in the need for antidiabetic, antihypertensive, and lipid-lowering medications [28]. Taken together, these observations indicate that any cardiometabolic advantage of RYGB is plausibly mediated by both greater, more durable weight loss and procedure-specific metabolic effects, cautioning against interpreting our non-significant pooled effects as therapeutic equivalence.

A hypocaloric diet reduces cardiovascular risk through convergent effects on lipids, blood pressure, glucose metabolism, inflammation, vascular structure, and myocardial biology. Caloric restriction improves the lipid profile by lowering LDL-cholesterol, ApoB, triglycerides, and the ApoB/ApoA-I ratio while increasing HDL-cholesterol and shifting lipoprotein composition toward less atherogenic particles [29]. It also reduces systolic and diastolic blood pressure via improved endothelial function, increased nitric oxide bioavailability through enhanced eNOS activity, and decreased oxidative stress and vascular inflammation [30]. Insulin sensitivity and glucose homeostasis improve through reductions in adiposity, better mitochondrial function, and favorable changes in metabolic intermediates, such as branched-chain amino acids and ketone bodies [31]. These effects are accompanied by lower circulating pro-inflammatory cytokines, attenuation of maladaptive vascular remodeling with preserved arterial compliance, improved mitochondrial content and reduced oxidative stress in the myocardium with anti-hypertrophic effects, and activation of nutrient-sensing pathways, including SIRT1 and AMPK, which enhance cellular stress resistance, autophagy, and metabolic efficiency [32,33].

RYGB engages many of these same pathways but also activates weight-loss-independent mechanisms that further improve cardiometabolic health. Beyond inducing substantial and sustained weight loss, RYGB produces marked reductions in LDL and non-HDL cholesterol, apolipoprotein B, and lipoprotein(a) via enhanced biliary cholesterol elimination, upregulation of hepatic bile acid-related genes, improved hepatic free-cholesterol handling, increased transintestinal cholesterol excretion, and reduced intestinal cholesterol absorption [34]. It improves endothelial function in a largely weight-independent manner through increased nitric oxide bioavailability and reduced oxidative stress while restoring HDL’s endothelial-protective, anti-inflammatory, antiapoptotic, and antioxidant functions [35]. RYGB decreases systemic inflammation, including interleukin-6, with parallel reductions in systolic blood pressure and downregulation of hepatic inflammatory genes [36]. Postoperative increases in incretin hormones (GLP-1, peptide YY) enhance insulin secretion and satiety, often preceding substantial weight loss [17,37,38], and alterations in bile acid pools and the gut microbiota further modulate lipid handling and insulin sensitivity [39,40]. Collectively, these mechanisms provide strong biological plausibility for the more profound and durable cardiometabolic benefits of RYGB compared with hypocaloric diets alone and help explain why short-term, tightly controlled studies may show similar intermediate changes while longer-term trajectories consistently favor surgical intervention.

Even with these advantages, hypocaloric diets and lifestyle modifications remain the cornerstone of obesity care and cardiovascular risk reduction worldwide [41]. Guidelines recommend energy restriction, physical activity, and behavioral therapy as first-line strategies because a 5–10% weight loss improves insulin sensitivity, blood pressure, and lipid profiles, lowering coronary and stroke risk [42,43,44]. Lifestyle change can also reduce systemic inflammation and improve endothelial function, offering additional vascular benefit [45]. The present findings—showing no statistically significant between-group differences for many outcomes—reinforce diet and lifestyle as foundational therapy, including when clinicians add surgery to optimize outcomes. In real-world settings, durable benefits from diet require structured programs and longitudinal behavioral support; without this infrastructure, recidivism and weight regain are common, which can attenuate long-term cardiometabolic improvement relative to surgical pathways.

Implementation of obesity treatments is highly unequal across health systems, especially between high-income countries and LMICs. New anti-obesity drugs (e.g., semaglutide, tirzepatide) achieve 15–20% weight loss and can reduce cardiovascular events, but they are best used as adjuncts to lifestyle therapy and, in selected cases, to surgery, rather than as stand-alone treatments [46,47]. However, in LMICs, limited funding, weak insurance coverage, scarce bariatric services, and under-resourced primary care restrict their use, while lifestyle and behavioral therapy remains the essential, lower-cost foundation of care—one that still requires multidisciplinary teams and intensive programs that are often not available [48].

This evidence base has limitations. Most comparative studies were observational or non-randomized, leaving room for selection bias and residual confounding [49,50,51]. Diet comparators varied (from prescribed low-calorie menus to general counseling), outcome timing differed, and follow-up was often short (<1–2 years), which may underestimate robust surgical effects and late complications. Although we prespecified a hypocaloric comparator to improve comparability, substantial clinical and methodological heterogeneity persisted. Safety reporting was inconsistent and underpowered to detect uncommon events; RYGB carries surgical and nutritional risks, while diet-only strategies face adherence barriers and metabolic compensation. These factors, together with small samples and high between-study variability, lower certainty and temper inferences.

Future work should prioritize long-term randomized comparisons (≥5–10 years) of RYGB against optimized non-surgical care that includes contemporary pharmacotherapy (e.g., GLP-1 receptor agonists), with endpoints that matter to patients: major cardiovascular events, kidney and microvascular outcomes, quality of life, and cost-effectiveness. Trials should standardize diet prescriptions, measurement methods, and safety definitions; capture adverse events and patient-reported outcomes systematically; and incorporate mechanistic substudies (e.g., lipoprotein(a), bile acids, incretin dynamics, microbiome) to identify targets that could inform surgical mimetics and precision treatment.

## 5. Conclusions

In this systematic review and meta-analysis, we did not find statistically significant differences between RYGB and hypocaloric diets for most cardiometabolic risk markers. Although point estimates tended to favor RYGB—particularly for BMI and triglycerides—the certainty of the evidence is very low, and follow-up in the included studies is predominantly short term (from weeks to about 12 months), with only sparse data beyond 1–2 years. Data on weight regain and on relapse of dyslipidemia, hypertension, and glycemic control were largely unavailable. Accordingly, our findings should be interpreted as describing short-term and intermediate-term changes in cardiometabolic markers, not long-term durability or relapse.

Within these constraints, hypocaloric diet and lifestyle modification remain the cornerstone of obesity care, while RYGB may provide additional short-term benefits in carefully selected patients who do not achieve adequate risk-factor control with optimized non-surgical treatment alone. Future comparative studies with longer follow-up and systematic assessment of weight regain, relapse of cardiometabolic risk factors, major cardiovascular events, quality of life, and cost-effectiveness are needed to define the long-term role of RYGB relative to modern non-surgical strategies.

## Figures and Tables

**Figure 1 jcm-14-08349-f001:**
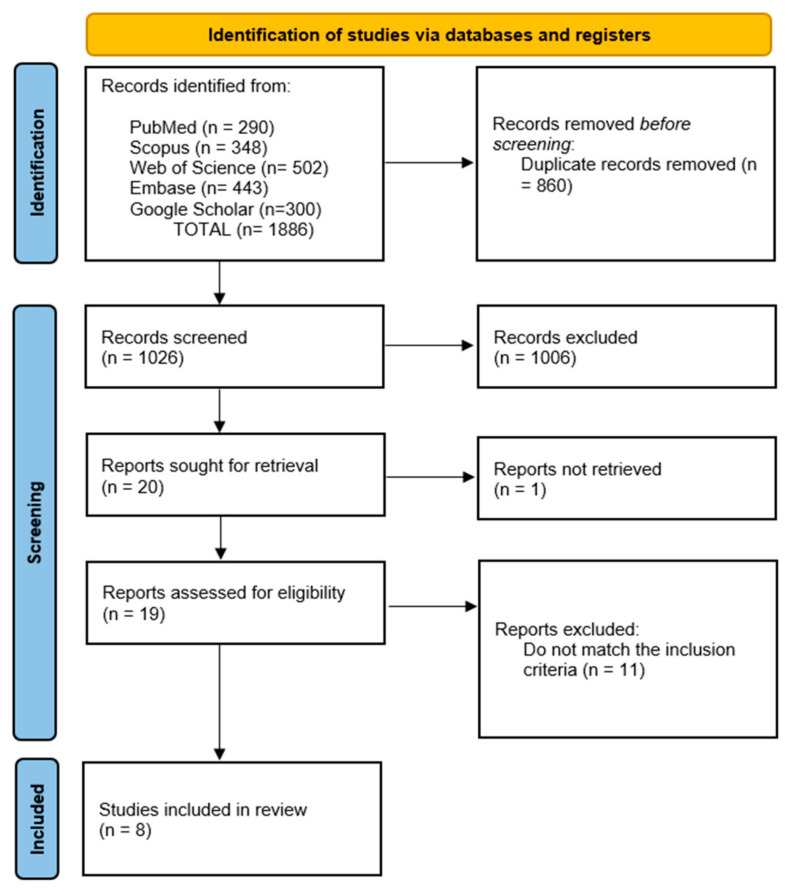
PRISMA flow diagram of study selection.

**Figure 2 jcm-14-08349-f002:**
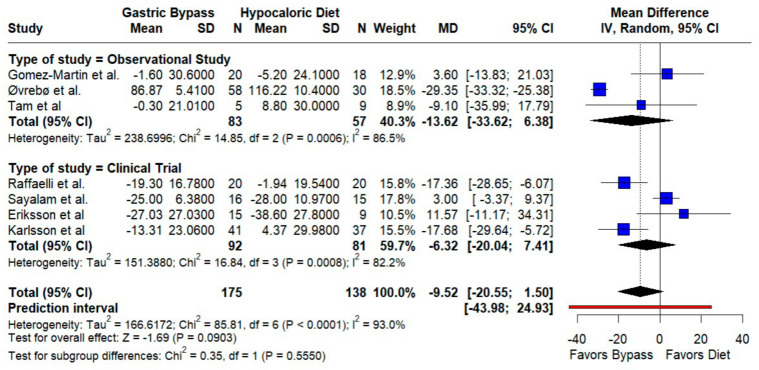
Forest plot showing the effect of gastric bypass surgery versus hypocaloric diets on lipid profiles. LDL cholesterol: Most studies trend toward lower LDL after RYGB versus hypocaloric diets; pooled estimates point in the same direction but remain imprecise, with marked between-study heterogeneity (high I^2^). Blue squares represent the mean difference for each individual study with 95% confidence intervals, black diamonds indicate the pooled random-effects estimates, and the red vertical line represents the line of no effect (MD = 0). Based on data from seven studies [10,16,17,18,19,20,21].

**Figure 3 jcm-14-08349-f003:**
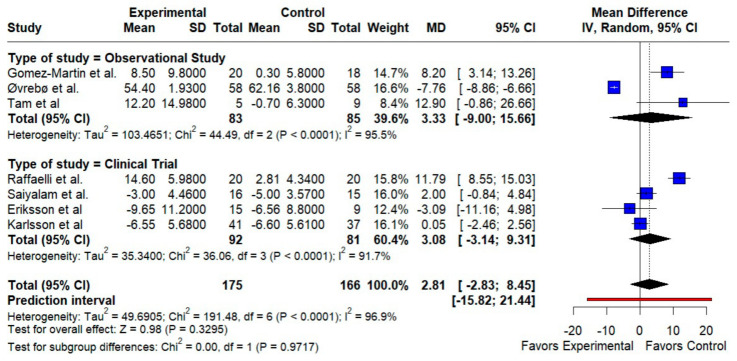
Forest plot showing the effect of gastric bypass surgery versus hypocaloric diets on lipid profiles. HDL cholesterol: RYGB shows a small, non-significant HDL increase overall. Observational and trial subgroups align in direction, but high heterogeneity limits confidence. Blue squares represent the mean difference for each individual study with 95% confidence intervals, black diamonds indicate the pooled random-effects estimates, and the red vertical line represents the line of no effect (MD = 0). Data are based on seven studies [10,16,17,18,19,20,21].

**Figure 4 jcm-14-08349-f004:**
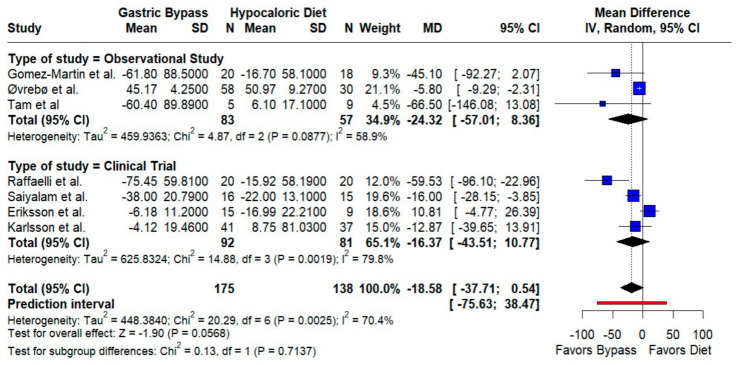
Forest plot showing the effect of gastric bypass surgery versus hypocaloric diets on lipid profiles. Triglycerides: Pooled effects favor RYGB with lower triglycerides; dispersion across studies is moderate to high, so the magnitude of benefit likely varies by setting. Blue squares represent the mean difference for each individual study with 95% confidence intervals, black diamonds indicate the pooled random-effects estimates, and the red vertical line represents the line of no effect (MD = 0). Data are based on seven studies [10,16,17,18,19,20,21].

**Figure 5 jcm-14-08349-f005:**
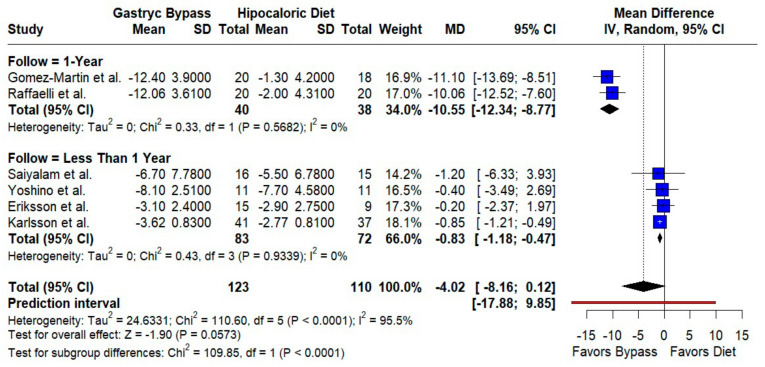
Forest plot comparing gastric bypass versus hypocaloric diet effects by follow-up time (1 year vs. less than 1 year) in body mass index. Negative mean differences (MDs) favor bypass. Subgroup analyses show low heterogeneity within groups but significant differences between follow-up durations (*p* < 0.0001). Blue squares represent the mean difference for each individual study with 95% confidence intervals, black diamonds indicate the pooled random-effects estimates, and the red vertical line represents the line of no effect (MD = 0). Data are based on six studies [10,16,17,18,19,22].

**Figure 6 jcm-14-08349-f006:**
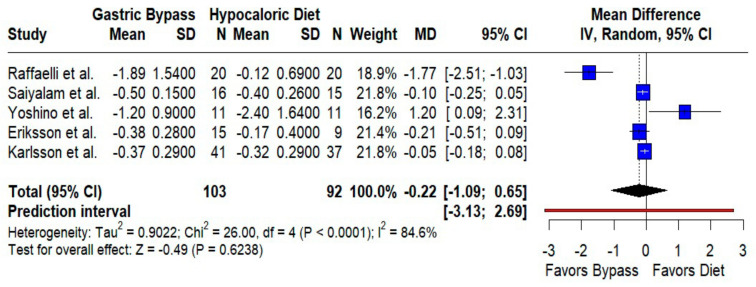
Forest plot comparing the effect of gastric bypass versus hypocaloric diets on HbA1c levels. Mean differences (MDs) with 95% confidence intervals (CIs) are shown for each study. Negative MDs favor gastric bypass. The overall effect shows no statistically significant difference (*p* = 0.62) with substantial heterogeneity (I^2^ = 84.6%). The prediction interval is included. Blue squares represent the mean difference for each individual study with 95% confidence intervals, black diamonds indicate the pooled random-effects estimate, and the red vertical line represents the line of no effect (MD = 0). Data are based on five studies [10,16,17,18,22].

**Figure 7 jcm-14-08349-f007:**
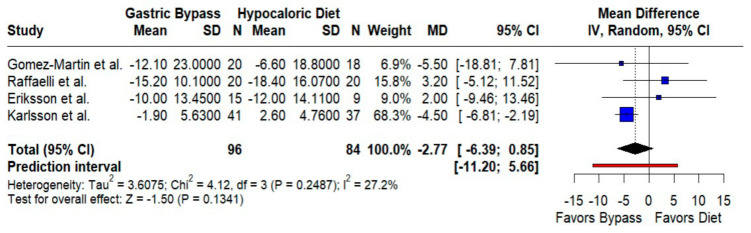
Forest plot comparing the effect of gastric bypass surgery versus hypocaloric diets on systolic blood pressure. Mean differences (MDs) with 95% confidence intervals (CIs) are shown for each study; negative MDs favor gastric bypass. The overall effect shows no statistically significant difference, with low between-study heterogeneity. Blue squares represent the mean difference for each individual study with 95% confidence intervals, black diamonds indicate the pooled random-effects estimate, and the red vertical line represents the line of no effect (MD = 0). Data are based on four studies [10,16,17,19].

**Table 1 jcm-14-08349-t001:** Characteristics of included studies comparing Roux-en-Y gastric bypass (RYGB) and hypocaloric diet interventions in obese adults. Summary of the key details of the eight studies included in the systematic review and meta-analyses. Abbreviations: SG, sleeve gastrectomy; RYGB, Roux-en-Y gastric bypass; VLED, Very Low Energy Diet; T2D, type 2 diabetes; HDL, high-density lipoprotein; LDL, low-density lipoprotein; TG, triglyceride; HOMA-IR, Homeostatic Model Assessment of Insulin Resistance; ApoA1, apolipoprotein A1; ApoB, apolipoprotein B; BP, blood pressure; HbA1c, glycated hemoglobin; PET/MRI, Positron Emission Tomography/Magnetic Resonance Imaging.

Author (Year)	Journal	Journal Ranking	Study Design	No. of Patients	Location; Period	Population	Intervention Type	Comparison	Main Outcomes
Eriksson et al. (2024) [16]	*Diabetologia*	Endocrinology and Metabolism, Q1	Randomized Clinical Trial	Surgery: 15; LCD: 9	Sweden; ~2022–2023	Obese adults without diabetes	Surgery vs. diet	SG/RYGB vs. Hypocaloric Diet (~1100 kcal/d, 4 weeks)	Glucose uptake (PET/MRI), HOMA-IR, lipids
Karlsson et al. (2024) [17]	*JAMA Surgery*	Surgery, Q1	Non-randomized Clinical Trial	RYGB: 41; VLED: 37	Norway; 2015–2017	Severe obesity, adults	Surgery vs. diet	RYGB vs. Hypocaloric Diet (<800 kcal/d, 6 weeks)	LDL, non-HDL, ApoB, Lp(a), glucose, BP
Saiyalam et al. (2024) [18]	*Nutrients*	Nutrition and Dietetics, Q1	Non-randomized Clinical Trial	SG: 89; RYGB: 89	Turkey; ~2021–2023	Obese with T2D	Surgical (SG vs. RYGB)	Surgical Hypocaloric Diet	HDL, LDL, TG, glucose, HbA1c
Raffaelli et al. (2014) [10]	*Annals of Surgery*	Surgery, Q1	Non-randomized Clinical Trial	RYGB: 40; Control: 40	Italy; ~2010–2012	Obese with metabolic syndrome	Surgery vs. diet/behavioral	RYGB vs. Hypocaloric Diet and Behavior Therapy	HDL, ApoA4, TG, HOMA-IR
Gómez-Martín et al. (2017) [19]	*Surgery for Obesity and Related Diseases*	Surgery, Q1	Observational Study	RYGB: 20; Control: 18	Spain; ~2015–2016	Obese females only	Surgery vs. control	Surgical vs. Hypocaloric Diet	IMT, endothelial dysfunction markers
Øvrebø B. et al. (2017) [20]	*Clinical Obesity*	Endocrinology, Diabetes, and Metabolism, Q2	Observational Study	SG: 67; RYGB: 67	Norway; ~2013–2015	Patients with morbid obesity	Surgical (SG vs. RYGB)	Surgical vs. Hypocaloric Diet	HDL, LDL, TG, ApoA1, ApoB, glucose, HbA1c
Tam et al. (2016) [21]	*Journal of Clinical Endocrinology and Metabolism*	Endocrinology and Metabolism, Q1	Observational Study	SG: 33; RYGB: 33	Hong Kong; ~2013–2015	Obese patients (BMI ≥ 35)	Surgical (SG vs. RYGB)	Surgical vs. Hypocaloric Diet	HDL, LDL, HOMA-IR, TG, glucose
Yoshino et al. (2020) [22]	*New England Journal of Medicine*	Medicine (miscellaneous), Q1	Observational Study	RYGB: 11; Diet: 11	USA; ~2019–2020	Obese patients	Surgery vs. diet	RYGB vs. Hypocaloric Diet 1000 kcal/d)	HDL, LDL, TG, glucose, HbA1c

## Data Availability

Data are contained within this article.

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
