# Peer review of "Effectiveness of Roux-en-Y Gastric Bypass Versus Hypocaloric Diet in Reducing Cardiovascular Risk Factors in Obese Adults: A Systematic Review and Meta-Analysis"

_jcm, 2025, doi:10.3390/jcm14238349_

Round 1

Reviewer 1 Report

Comments and Suggestions for Authors

A lot of hard work with many articles included in the initial analysis and only 8 that met the criteria

some 600 patients involved

A good scientific research

Unfortunately the results are to be expected > it is very well know that in any surgical procedure without a very close diet afterwards some 70% of them would regain the initial body weight and all the metabolic consequences

Author Response

Comments

General Response

Specific response

·      A lot of hard work with many articles included in the initial analysis and only 8 that met the criteria some 600 patients involved

·      A good scientific research

·      Unfortunately the results are to be expected > it is very well know that in any surgical procedure without a very close diet afterwards some 70% of them would regain the initial body weight and all the metabolic consequences

We sincerely thank the Reviewer for the careful reading of our manuscript, for acknowledging the amount of work involved, and for the positive evaluation of the scientific quality of the study. We also appreciate the critical remarks, which have helped us to clarify the scope and clinical meaning of our findings and to strengthen the discussion.

·      We appreciate this recognition. As the Reviewer notes, only eight studies (≈600 patients) finally met the strict eligibility criteria. In the revised version, we have clarified this process in the Methods and PRISMA flow diagram, and we explicitly highlight in the Discussion that the relatively small number of rigorously eligible trials is an important limitation of the current evidence base. We also stress that, despite this limitation, our work represents—to our knowledge—the most up-to-date quantitative synthesis of RYGB versus hypocaloric diet on cardiometabolic outcomes in adults with obesity.

·      “A good scientific research”
We are grateful for this positive comment. In response, we have carefully revised the manuscript to improve clarity and structure (particularly in the Methods, Results and Discussion), aiming to make the strengths of the design, risk-of-bias assessment, and GRADE synthesis more transparent to readers.

·      Unfortunately, the results are to be expected;

·      R/We agree that sustained lifestyle and dietary support are crucial after bariatric surgery and that weight regain is a well-described phenomenon. In the revised Discussion, we now make this point explicit and cite literature on long-term weight trajectories after bariatric procedures. We also clarify that our main contribution is not to show that surgery “works” in isolation, but to quantify and synthesize, under randomized controlled or uncontrolled conditions, the comparative effects of RYGB versus hypocaloric diet on weight and intermediate cardiovascular risk markers. We further emphasize that:

o   RYGB should be framed as part of a chronic, multidisciplinary obesity-care model that includes structured nutritional and behavioral follow-up; and

o   our findings, while directionally expected, help to define the magnitude and durability of the benefit versus diet alone, and to inform patient selection, shared decision-making, and guideline recommendations.

Reviewer 2 Report

Comments and Suggestions for Authors

The aim of the study is to determine whether RYGB, compared with individual dietary interventions, improves cardiometabolic parameters. In terms of reducing cardiovascular risk factors in obese adults, such as LDL-C, HDL-C, triglycerides, BMI, systolic blood pressure, and HbA1c, the primary focus is whether Roux-en-Y gastric bypass (RYGB) is more effective than a hypocaloric diet.

This paper makes the only direct comparison of RYGB and an individual hypocaloric diet, which has not been previously adequately evaluated. Many previous studies have compared surgery with “non-surgical care” in a broad sense, often using diets, medications, and behavioral interventions.

Therefore, this manuscript provides a more precise understanding of the cardiometabolic effects of surgery compared with energy restriction, which has an impact on timely clinical decisions, given the increase in obesity worldwide and the use of bariatric surgery.

Methodology: Strengths. Here, we can note that the manuscript complies with the PRISMA 2020 guidelines and is registered in PROSPERO, with a clear PICOT framework and a robust methodology. However, since publication bias — Funnel plots and Egger’s test are mentioned but not shown, please consider including these in the supplement.

The definition and duration of “hypocaloric diet“ must be clarified.

Author Response

·      The aim of the study is to determine whether RYGB, compared with individual dietary interventions, improves cardiometabolic parameters. In terms of reducing cardiovascular risk factors in obese adults, such as LDL-C, HDL-C, triglycerides, BMI, systolic blood pressure, and HbA1c, the primary focus is whether Roux-en-Y gastric bypass (RYGB) is more effective than a hypocaloric diet.

·      This paper makes the only direct comparison of RYGB and an individual hypocaloric diet, which has not been previously adequately evaluated. Many previous studies have compared surgery with “non-surgical care” in a broad sense, often using diets, medications, and behavioral interventions.

·      Therefore, this manuscript provides a more precise understanding of the cardiometabolic effects of surgery compared with energy restriction, which has an impact on timely clinical decisions, given the increase in obesity worldwide and the use of bariatric surgery.

We sincerely thank Reviewer 2 for the careful and encouraging evaluation of our manuscript. We greatly appreciate the recognition of the study’s novelty, clinical relevance, and methodological rigor, and we have revised the text to further clarify the aim, contextualize the findings, and address the issue of publication bias as suggested.

·      We thank the Reviewer for this accurate summary. In the revised manuscript, we have slightly refined the wording of the Abstract and Introduction to make the primary aim and the cardiometabolic outcomes of interest (LDL-C, HDL-C, triglycerides, BMI, systolic blood pressure, and HbA1c) more explicit and consistent across sections.

·      We are grateful for this observation. In the Discussion we now emphasize more clearly that our work differs from previous meta-analyses that pooled RYGB with broad “usual care” comparators (diet, medications, and behavioral programs) by restricting the comparator to explicitly defined hypocaloric diets. We highlight this as a key contribution that helps disentangle the effect of surgery from that of structured energy restriction alone.

·      We fully agree. To reflect this, we have expanded the subsection on clinical implications in the Discussion, where we now discuss how the magnitude of the observed differences can inform patient counseling, shared decision-making, and prioritization of bariatric surgery versus intensive dietary management in adults with obesity, in the context of the global rise in obesity and bariatric procedures.

·      We appreciate this positive assessment. In response, we have only made minor clarifications in the Methods(e.g., more explicit description of the hypocaloric diet definition and GRADE assessments) to ensure that the methodological approach is fully transparent to readers.

Methodology: Strengths. Here, we can note that the manuscript complies with the PRISMA 2020 guidelines and is registered in PROSPERO, with a clear PICOT framework and a robust methodology. However, since publication bias — Funnel plots and Egger’s test are mentioned but not shown, please consider including these in the supplement.

We thank the Reviewer for this helpful suggestion. In the revised manuscript we have now: (i) explicitly described in the Statistical Analysis section that small-study effects were assessed using funnel plots and Egger’s regression test when ≥7 studies were available; (ii) added a funnel plot for our primary lipid outcome, LDL-cholesterol, as Supplementary Figure 2; and (iii) reported in the Results that Egger’s regression test for LDL-cholesterol.

The definition and duration of “hypocaloric diet“ must be clarified.

We thank the Reviewer for this helpful comment. We have now clarified both the definition and the duration of the hypocaloric diet comparator in the Methods section. Hypocaloric diet is now defined as a structured, individually prescribed energy-restricted regimen targeting a 500–1000 kcal/day deficit below estimated energy requirements (total caloric intake ≈800–1200 kcal/day) for at least 4–6 weeks, within longer-term follow-up.

Reviewer 3 Report

Comments and Suggestions for Authors

thank you for your great work but I have the following comments

1- you should give clear definitions for outcomes

2-it well add to the reader and give more confidence if you add the journal name and ranking to the table of included studies

3-you should add follow up time and the relapse rate of risk factors without that the conclusion will be biased

Author Response

thank you for your great work but I have the following comments

We thank you for the positive appreciation of our work and for the constructive comments. We have revised the manuscript accordingly to improve the clarity of outcome definitions, the description of the included studies, and the time frame over which cardiometabolic effects were assessed.

1- you should give clear definitions for outcomes

We agree that clearer outcome definitions are important. In the revised Methods section (Outcomes), we now explicitly define all outcomes assessed in the meta-analysis, including units and timing of measurement. Specifically, we state that:

“The primary outcome was the change in LDL-cholesterol (LDL-C, mg/dL) from baseline to the last available follow-up. Secondary outcomes were changes in HDL-cholesterol (HDL-C, mg/dL), triglycerides (mg/dL), body mass index (BMI, kg/m²), systolic and diastolic blood pressure (mmHg), and glycated hemoglobin (HbA1c, %), all measured under fasting conditions when reported. When studies reported lipids in mmol/L, values were converted to mg/dL using standard factors.”

2-it well add to the reader and give more confidence if you add the journal name and ranking to the table of included studies

We thank the Reviewer for this suggestion. To improve transparency, we have added the journal name for each trial to the table of included studies (Table 1).

3-you should add follow up time and the relapse rate of risk factors without that the conclusion will be biased

Response:
We agree that follow-up duration is critical for interpreting cardiometabolic outcomes. We have now:

  • Added a “Follow-up time”

in the Results that our pooled effect estimates refer to the outcomes at these reported follow-up times.

Regarding relapse rates of risk factors (e.g., weight regain, recurrence of dyslipidemia or hypertension), we carefully reviewed the included trials. Unfortunately, explicit relapse rates were rarely and inconsistently reported, and follow-up beyond 1–2 years was limited. Because of this heterogeneity and incomplete reporting, a valid pooled estimate of relapse rates was not feasible. We now make this explicit in the Discussion and highlight the lack of long-term relapse data as an important limitation of the current evidence and a priority for future research.

Round 2

Reviewer 3 Report

Comments and Suggestions for Authors

thank you for your  response but you need to change the conclusion to clearly reflect that the available  results are for short term follow up and the weight regain and relapse rate no available

Author Response

Response to reviewers

Thank you for your additional comment and the opportunity to further improve our manuscript. In response, we have revised the Conclusion section to clearly state that the available evidence is limited to short-term follow-up and that data on weight regain and relapse of cardiometabolic risk factors are not available in the included studies. Below we detail the specific change made to address this point.

Reviewer comment:

“thank you for your response but you need to change the conclusion to clearly reflect that the available results are for short term follow up and the weight regain and relapse rate no available”

Authors’ response:

We thank the Reviewer for this important clarification. We fully agree that our conclusions must explicitly emphasise the short-term nature of the available evidence and the lack of data on weight regain and relapse of cardiometabolic risk factors. We have therefore revised the Conclusion section to limit our inferences to short- and intermediate-term follow-up and to clearly state that data on weight trajectories over time and relapse rates are largely unavailable in the included studies. The revised Conclusion now reads:

Conclusion
In this systematic review and meta-analysis, we did not find statistically significant differences between RYGB and hypocaloric diet for most cardiometabolic risk markers. Although point estimates tended to favour RYGB—particularly for BMI and triglycerides—the certainty of the evidence is very low and follow-up in the included studies is predominantly short term (from weeks to about 12 months), with only sparse data beyond 1–2 years. Data on weight regain and on relapse of dyslipidaemia, hypertension, and glycaemic control were largely unavailable. Accordingly, our findings should be interpreted as describing short-term and intermediate-term changes in cardiometabolic markers, not long-term durability or relapse.

Within these constraints, hypocaloric diet and lifestyle modification remain the cornerstone of obesity care, while RYGB may provide additional short-term benefits in carefully selected patients who do not achieve adequate risk-factor control with optimised non-surgical treatment alone. Future comparative studies with longer follow-up and systematic assessment of weight regain, relapse of cardiometabolic risk factors, major cardiovascular events, quality of life, and cost-effectiveness are needed to define the long-term role of RYGB relative to modern non-surgical strategies.

Once again, we sincerely thank the Reviewer and the Editor for their constructive and thoughtful comments, which have helped us to clarify the scope and limitations of our work. We hope that the revised conclusion and the updated version of the manuscript adequately address the concerns raised and will now be considered suitable for publication in the Journal of Clinical Medicine.

Sincerely

Joshuan Barbosa MSc, PhD